# Quantitative Gait Feature Assessment on Two-Dimensional Body Axis Projection Planes Converted from Three-Dimensional Coordinates Estimated with a Deep Learning Smartphone App

**DOI:** 10.3390/s23020617

**Published:** 2023-01-05

**Authors:** Shigeki Yamada, Yukihiko Aoyagi, Chifumi Iseki, Toshiyuki Kondo, Yoshiyuki Kobayashi, Shigeo Ueda, Keisuke Mori, Tadanori Fukami, Motoki Tanikawa, Mitsuhito Mase, Minoru Hoshimaru, Masatsune Ishikawa, Yasuyuki Ohta

**Affiliations:** 1Department of Neurosurgery, Nagoya City University Graduate School of Medical Science, Nagoya 467-8601, Japan; 2Normal Pressure Hydrocephalus Center, Rakuwakai Otowa Hospital, Kyoto 607-8062, Japan; 3Interfaculty Initiative in Information Studies/Institute of Industrial Science, The University of Tokyo, Tokyo 113-8654, Japan; 4Digital Standard Co., Ltd., Osaka 536-0013, Japan; 5Division of Neurology and Clinical Neuroscience, Department of Internal Medicine III, Yamagata University School of Medicine, Yamagata 990-9585, Japan; 6Human Augmentation Research Center, National Institute of Advanced Industrial Science and Technology (AIST), Kashiwa II Campus, University of Tokyo, Kashiwa 277-0882, Japan; 7Shin-Aikai Spine Center, Katano Hospital, Katano 576-0043, Japan; 8School of Medicine, Shiga University of Medical Science, Otsu 520-2192, Japan; 9Department of Informatics, Faculty of Engineering, Yamagata University, Yamagata 992-8510, Japan; 10Rakuwa Villa Ilios, Rakuwakai Healthcare System, Kyoto 607-8062, Japan

**Keywords:** deep learning, motion tracking, markerless motion capture, quantitative gait assessment, smartphone device

## Abstract

To assess pathological gaits quantitatively, three-dimensional coordinates estimated with a deep learning model were converted into body axis plane projections. First, 15 healthy volunteers performed four gait patterns; that is, normal, shuffling, short-stepped, and wide-based gaits, with the Three-Dimensional Pose Tracker for Gait Test (TDPT-GT) application. Second, gaits of 47 patients with idiopathic normal pressure hydrocephalus (iNPH) and 92 healthy elderly individuals in the Takahata cohort were assessed with the TDPT-GT. Two-dimensional relative coordinates were calculated from the three-dimensional coordinates by projecting the sagittal, coronal, and axial planes. Indices of the two-dimensional relative coordinates associated with a pathological gait were comprehensively explored. The candidate indices for the shuffling gait were the angle range of the hip joint < 30° and relative vertical amplitude of the heel < 0.1 on the sagittal projection plane. For the short-stepped gait, the angle range of the knee joint < 45° on the sagittal projection plane was a candidate index. The candidate index for the wide-based gait was the leg outward shift > 0.1 on the axial projection plane. In conclusion, the two-dimensional coordinates on the body axis projection planes calculated from the 3D relative coordinates estimated by the TDPT-GT application enabled the quantification of pathological gait features.

## 1. Introduction

A pathological gait is a gait that deviates from a normal gait, and its features are characterized as shuffling, short-stepped, freezing, wide-based, festination, hemiplegic, spastic, ataxic, and instability. [1,2,3,4,5,6]. Although videorecording has been clinically recommended when evaluating pathological gaits [2,3,4,5], these gait features have been evaluated subjectively without any standardized rating systems. The qualitative or semi-quantitative subjective assessments are simple and widely used in clinical practice, including medical record input, but their validity and reliability are generally low [1,2]. Therefore, gait analysis studies to assess movement disorders require quantitative values or indices using some type of instrumentation; for example, a reduced stride length and diminished step height were reported to be typical spatiotemporal and kinematic characteristics of a pathological gait in Parkinson’s disease and idiopathic normal pressure hydrocephalus (iNPH) [6,7,8,9,10,11,12], asymmetry of step length, and leg swing velocity were typical for a hemiplegic gait [13,14], and coefficients of variation of stride length and step width were typical for an ataxic gait and instability [15]. The three-dimensional (3D) acceleration of the trunk has been reported to be a useful parameter for detecting pathological gait features. In our previous study, however, 3D trunk acceleration during straight walking could not clearly distinguish the six gait features of shuffling, short-stepped, freezing, wide-based, festination, and instability in patients with iNPH [16]. Therefore, we reaffirmed the need to analyze the spatiotemporal and kinematic parameters of gait; for instance, the joint angle, stride length, and step width using a motion capture technology. An optical 3D motion capture system with multipoint cameras and numerous body markers is a precise method for gait analysis and has become the gold standard [17,18,19,20]. However, it has not been clinically applied because it takes time and effort to train technicians and carry out measurements and analysis, and the equipment is expensive. Therefore, we developed a novel iOS application named Three-Dimensional Pose Tracker for Gait Test (TDPT-GT) (https://digital-standard.com/tdptios_en/, accessed on 1 December 2022) in which the 3D motion of the full body was simply tracked from two-dimensional (2D) video images captured by a single iPhone camera [21]. This application can estimate the 3D relative coordinates of 24 key points in a full body-pose during movement without time-consuming complicated preparation. Furthermore, we developed a new method for assessing pathological gaits more easily by converting 3D relative coordinates into 2D coordinates projected on the sagittal, coronal, and axial planes with respect to the body axis. In this study, we explored the novel distinctive indices that express the essence of each pathological gait feature from 2D relative coordinates projected on the sagittal and axial planes.

## 2. Materials and Methods

### 2.1. Ethical Approvals

The study design and protocol of this study were approved by the ethics committee for human research at our institute (IRB number: R2019-337). All volunteers and patients participated in this study after providing written informed consent. The study design was prospective and observational. This study was conducted according to the approved guidelines of the Declaration of Helsinki.

### 2.2. Study Population

From November 2020 to August 2022, 15 healthy medical students, medical staff, and their families, aged 20 years or older, were recruited by open recruitment. The inclusion criteria for this study were independence in their daily lives, without any problems in walking, normal cognitive function, and no history of neurodegenerative disorders, severe head injury, or cerebrovascular disease. The volunteers walked normally and acted out pathological gait features, such as a short-stepped gait, short-stepped shuffling wide-based gait, and short-stepped shuffling closed gait. A short-stepped gait was defined as a reduced stride length, and a shuffling gait had a diminished step height. In addition, we acquired two additional datasets to assess the generalizability of the candidate parameters and indices. The first dataset contained data from 47 patients diagnosed with iNPH, and the second dataset contained data from 92 healthy volunteers aged 60 years or older who participated in the Takahata population-based cohort study [22,23,24]. The inclusion criteria for all participants, including iNPH patients, were independence in their ability to walk without any assistance and the ability to make independent decisions. These patients participated in the study on their own initiative. The patients with iNPH were subjectively evaluated for the presence or absence of the following six pathological gait features: a shuffling gait, a short-stepped gait, a wide-based gait, a freezing gait, a spastic gait, and instability [16]. The simulated pathological gaits of the healthy volunteers were quite extreme, compared with those of the patients with iNPH.

### 2.3. Data Acquisition of Estimated Three-Dimensional Relative Coordinates

The walking movements of participants while walking in a circle with a diameter of 1 m for two laps, clockwise and counterclockwise, were recorded from head to toe in the videoframes of the TDPT-GT application. The TDPT-GT application worked precisely under the following conditions. The lines of the subject’s body and legs were as easy to identify as possible. The recording took place in a bright space with a uniform background where no mirror was present and no one other than the subject appeared in the video frame. Details of the technology of the TDPT-GT application and data acquisition were described in our prior publication [21]. In brief, the TDPT-GT application is an artificial intelligence (AI)-based motion capture system using a 3D heatmap estimation that could measure the 3D relative coordinates of the human body at 30 fps with 448 × 448 pixels and RGB color using an iPhone camera without any markers for motion capture. The TDPT-GT application estimates the 3D relative coordinates of the following 24 key points: the center of the body (navel), nose, left and right ears, eyes, shoulders, elbows, wrists, thumbs, middle fingers, hips, knees, heels, and toes, based on the center of the body, not the absolute coordinates that defined the ground. The x- and y-axis coordinates on the screen were converted directly into the three-dimensional x- and y-axis coordinates, and the center point of depth on the screen was always set at the subject’s navel. The raw and smoothed coordinates were applied to the low-pass filter, and the AI scores, which represented the certainty and probability of coordinate estimations, were automatically calculated and saved as a csv file format in the iPhone. The head center was defined as the center coordinates of both ears, and the neck center was defined as the center coordinates of both shoulders. For relative 3D coordinates, the length of the upper body was calculated as the length from the navel center to the neck center and the length from the neck center to the head center, and this measurement was set as 1. The leg length was calculated as the total length from the center to the hip joint, from the hip joint to the knee, and from the knee to the heel, and this measurement was set as 1. The reliability of the 3D relative coordinates estimated by the TDPT-GT application was evaluated and compared with 3D optical motion capture systems, described in a previous study [21].

### 2.4. Projection of Relative Coordinates on the Body Axis Planes

As shown in Figure 1, 2D relative coordinates of the 24 key points were calculated from 3D relative coordinates with an AI score of 0.7 or higher and projected onto the sagittal, coronal (frontal), and axial planes based on the body axis. The normal vector to a plane passing through the navel center and right and left shoulder points was defined as $\vec{UF}$ and the normal vector to a plane passing through the navel center and right and left hip joints was defined as $\vec{DF}$. The $\vec{f}$ for the forward direction of the body axis was calculated as follows: $\left(\vec{UF}+\vec{DF}\right)/2$. A sagittal plane was defined as a plane composed of $\vec{f}$ and $\vec{c}$ from the navel center to the center between the bilateral hip joints. A coronal (frontal) plane was defined as a plane composed of the $\vec{c}$ and $\vec{n}$ that was orthogonal to the two vectors of $\vec{f}$ and $\vec{c}$. An axial plane was defined as a plane composed of $\vec{f}$ and $\vec{n}$; that is, $\vec{f}$·$\vec{c}$ = $\vec{f}$·$\vec{n}$ = $\vec{n}$·$\vec{c}$ = 0. Each 2D coordinate projected on the sagittal, coronal, and axial planes of each 3D relative coordinate was the intersection of the projected plane and the normal line with respect to the projected plane. In addition, the 2D relative coordinates of the bilateral hip joints, knees, heels, and toes were corrected by the total leg length as 1 on the projected plane, and those of the upper body points were also corrected by the total length of the upper body as 1.

Estimated three-dimensional coordinates of 24 full body key points were converted to the two-dimensional relative coordinates by the projection onto the sagittal, coronal, and axial planes based on the body axis. First, the forward vector $\vec{f}$ relative to the body axis from the center and both shoulder points (light brown) and hip joints (light salmon) was determined. Next, the downward vector $\vec{c}$ relative to the body axis from the center and both hip joints was determined.

### 2.5. Data Processing for Gait Analysis

The chronological changes in the 2D relative coordinates excluding the upper limbs were plotted on the sagittal, coronal (frontal), and axial planes, as shown in Figure 2. Additionally, 75% tolerance ellipses of all plots for each key point were drawn on the projection planes. The elliptical area, center coordinate of the ellipses, tilt angles of major and minor axes, and the x-axis and y-axis lengths of the 75% tolerance ellipses were calculated at each 2D relative coordinate on the projection planes as follow: dataEllipse (x coordinates of key point, y coordinates of key point, levels = 0.75). The center coordinate of the ellipse was calculated as cov.wt (cbind (x coordinates of key point, y coordinates of key point)), and the elliptical area was calculated as the length of the major axis of the ellipse × the length of the minor axis of the ellipse × π (pi).

On the sagittal projection plane (Figure 2, lower left), an angle range of the hip joint was defined as an angle between two vectors from the center coordinate of the 75% tolerance ellipse for the hip joint to two coordinates of the major axis for the knee ellipse, and an angle range of the knee joint was defined as an angle between two vectors from the knee ellipse center to two coordinates of the major axis for the heel ellipse. A relative vertical amplitude of the heel was defined as the y-axis width of the heel plots.

On the axial projection plane (Figure 2, lower right), a foot angle [6], i.e., outward rotation of the feet, was defined as an angle between two vectors from the right and left heel ellipse centers to the right and left toe ellipse centers, respectively. A toe outward shift was defined as a lateral deviation of the right and left x-coordinates of toe ellipse centers from those of the heel ellipse centers, a heel outward shift was defined as a lateral deviation of the right and left x-coordinates of the heel ellipse centers from those of the hip ellipse centers, and a leg outward shift was defined as the sum of the toe and heel outward shifts. The indices of the 2D relative coordinates projected onto the coronal (frontal) plane were not investigated in this study because the bending direction of the knee joint or ankle joint was sagittal, and the foot angle or outward shift was easier to assess on the axial plane rather than on the coronal plane.

### 2.6. Statistical Analysis

After detecting appropriate indices for each pathological gait feature by the first comprehensive parameter search in the gait dataset of 15 healthy volunteers, optimal cutoff values of the candidate indices were investigated based on the distribution of the measurements and indices in the iNPH patients with gait disturbances and healthy individuals aged 60 years or older in the Takahata cohort study. In addition, the detectability of the shuffling gait, short-stepped gait, and wide-based gait at the setting cutoff values was evaluated with the area under the receiver-operating characteristic curve (AUC), sensitivity, specificity, and odds ratio. The statistical significance was assumed at a probability (*p*) value of <0.05. All missing data points were treated as deficit data that did not affect other variables. Statistical analyses were performed using R (version 4.1.1; The R Foundation for Statistical Computing; http://www.R-project.org accessed on 1 December 2022).

## 3. Results

### 3.1. Clinical Characteristics

Fifteen healthy volunteers (mean age 39.1 ± 20.1 years; range, 22–78 years; nine males and six females; height 148–194 cm) were assessed for a normal gait and the acting pathological gait features as follows: short-stepped shuffling wide-based gait, short-stepped shuffling closed gait, and short-stepped gait, according to the TDPT-GT application. As the second step, the gait of 47 patients diagnosed with iNPH (mean age 77.3 ± 6.3 years; range, 61–87 years; 32 males and 15 females) and 92 healthy elderly individuals (mean age 73.0 ± 6.3 years; range, 60–91 years; 36 males and 56 females) in the Takahata cohort study were assessed using the TDPT-GT. Table 1 summarizes the participants’ characteristics.

### 3.2. Two-Dimensional Relative Coordinates on the Sagittal and Axial Projection Planes

The movement trajectories (plot distribution) and 75% tolerance ellipse parameters for the 2D relative coordinates projected onto the sagittal and axial planes were compared between the normal gaits and acting pathological gaits in the first 15 healthy volunteer group (Figure 3).

Compared to the normal gait, the range of the back-and-forth leg swing was apparently smaller in the short-stepped shuffling gait, regardless of whether the legs were opened or closed. In the short-stepped but non-shuffling gait (Figure 3b), the range of the back-and-forth swing of the lower legs, i.e., the heels and toes, was also smaller because of the smaller stride length, but the knees moved markedly in an upward and forward direction.

#### 3.2.1. Sagittal Projection Plane

The shuffling gait could be clearly distinguished by the angle range of the hip joint, angle range of the knee joint, and relative vertical amplitude of the heel (Figure 4, Figure 5 and Figure 6), whereas the short-stepped gait could be clearly distinguished only by the angle range of the knee joint (Figure 5). As the optimal cutoff values, <30° of the angle range of the hip joint for detecting a shuffling gait (Figure 4) and <45° of the angle range of the knee joint for detecting a shuffling gait and short-stepped gait (Figure 5) were proposed. As the optimal cutoff value for detecting the shuffling gait, we proposed <0.1 of the relative vertical amplitude of the heel with the total leg length set as 1 on the sagittal projection plane (Figure 6).

#### 3.2.2. Axial Projection Plane

On the axial projection plane, the foot angle between the right and left vectors from the heel to toes could not distinguish the wide-based and normal gaits (Figure 7).

The wide-based gait could be clearly distinguished by the heel outward shift, toe outward shift, and leg outward shift, which was the sum of the heel and toe outward shift (Figure 8, Figure 9 and Figure 10). As the optimal cutoff values for a wide-based gait, we proposed >−0.08 of the heel outward shift (Figure 8), >0.18 of the toe outward shift (Figure 9), and >0.1 of the leg outward shift, with the total leg length set as 1 on the axial projection plane (Figure 10).

### 3.3. Discrimination of Pathological Gaits by Indices on 2D Projection Planes Using the TDPT-GT App

Although we subjectively evaluated shuffling and short-stepped and wide-based gaits in 47 patients with iNPH, as shown in Table 1, these gait types were not used for investigating indices of 2D relative coordinates projected onto the sagittal or axial planes of the body axis, because the validity of subjective assessments of gait features is known to be low [2] and the shuffling, short-stepped, and wide-based gait are closely related to each other [16]. Instead, the AUC (95% CIs), sensitivity, specificity, and odds ratio (95% CIs) at the setting cutoff values were evaluated (Table 2). Of them, <45° of the angle range of the knee joint for detecting a short-stepped gait had the highest AUC (81.7, 95% CIs = 77.4–86.0) and the highest odds ratio (10.55, 95% CIs = 5.63–21.12).

## 4. Discussion

To establish generalized indices for pathological gaits that can be commonly used, it is necessary to first extract the feature quantities of pathological gaits using a simple method and then compare the distribution of them between healthy and affected subjects to determine the optimal cutoff values. Prior to this study, we developed a novel markerless monocular motion capture smartphone application that could obtain the moving 3D relative coordinates of the entire human body based on simple 2D video recordings [21]. In this study, we suggested that the angle range of the hip joint be <30° and the relative vertical amplitude of the heel < 0.1 as novel indices for detecting a shuffling gait, and the angle range of the knee joint < 45° as a useful index for detecting a shuffling gait and short-stepped gait on the sagittal projection plane calculated from the 3D relative coordinates, estimated by the TDPT-GT application. In addition, to detect a wide-based gait, a leg outward shift > 0.1 on the axial projection plane was proposed as a useful index rather than the foot angle. The most characteristic gait feature specific to iNPH to differentiate it from Parkinson’s disease was suggested to be an opening foot angle, i.e., an outward rotation of the feet [6]. The reason why our result was inconsistent with previous reports could be that our task included walking around a circle with a diameter of 1 m in order to hold the smartphone close to the examinee. Using the TDPT-GT application with our method, it is possible to assess gait in a confined space, such as a private clinic or one’s own room. However, toes are likely to be open with a normal gait and the foot angle is not a useful index for detecting a wide-based gait with our measurement method.

In general, a person’s normal gait changes with age. Compared to younger adults, older adults walk slowly with a shorter and wider step due to decreased forward propulsion [19,20,25,26,27]. Therefore, we needed to evaluate the gait of patients with iNPH and age-matched controls using the same method to determine candidate indices and their cutoff values for detecting pathological gait features. Furthermore, we needed to evaluate simulated pathological gaits in volunteers, separate from the case-controlled group, to ensure the identification of the pathological gait features of the shuffling, short-stepped, and wide-based gaits because these gait patterns are closely related to each other and could not be clearly distinguished in the real patients with iNPH and older adults. For example, a wide-based gait reduces the stride length, resulting in a short-stepped gait and a shuffling gait. Conversely, a shuffling gait with large steps or a short-stepped gait with rising legs is not a normal situation when walking.

The greatest advantage of this study is that any clinician or researcher can easily evaluate a pathological gait using a simple smartphone application. Optical 3D motion capture systems with multipoint cameras and markers for motion capture are prohibitively expensive and may be unsuitable for general use. In contrast, our proposed gait assessment can be applied easily without considerable amounts of time and money. Furthermore, in our system, large amounts of human motion data can be processed within a few milliseconds. This enables the application to perform in real time to support capturing the 3D pose, analyzing gait, and encouraging physical exercise. Consequently, these measurements can be used in multicenter collaborative studies to assess the severity of the pathological gait and changes in gait disturbances using a monocular smartphone camera.

This study has some limitations that warrant discussion. First, the 2D relative coordinates projected onto the sagittal or axial planes of the body axis based on the navel as the center without the ground information could not be compared with the previous conventional gait research, which was usually assessed by separating the stance phase from the swing phase or measuring the distance from the floor surface. However, the gait cycle time, stride length, step frequency, and angle range of the joint can be estimated easily using our method. Second, this system could track the body pose of one person only, and the iPhone camera must always show the entire body. Therefore, it may not be usable in a cluttered environment, such as a group health check. In order to implement our simple gait assessment in society using the TDPT-GT application, it is necessary to accumulate experience in more collaborate studies and clarify its advantages and disadvantages in our proposed gait analysis without specifying the floor surface. The development of objective measurements would greatly help with assessing the efficacy of therapeutic interventions, including medications, surgery, and the rehabilitation for any movement disorders.

## 5. Conclusions

The 2D coordinates projected on the sagittal, coronal, and axial planes with respect to the body axis were calculated from the 3D relative coordinates estimated by our AI-based iPhone application. Using the TDPT-GT application, anyone can easily evaluate various pathological gait patterns quantitatively, which previously required a large-scale 3D motion analysis system. The candidate indices for the shuffling gait were the angle range of the hip joint < 30°, the angle range of the knee joint < 45°, and relative vertical amplitude of the heel < 0.1 on the sagittal projection plane. The candidate index for the short-stepped gait was the angle range of the knee joint < 45° on the sagittal projection plane. The candidate index for the wide-based gait was not the foot angle but the leg outward shift > 0.1 on the axial projection plane. In the future, this AI-based application will quantify pathological gaits, making it possible for non-specialist individuals to easily detect fall risks and gait disorders at an early stage, which will help to prevent worse conditions that would require nursing care.

## Figures and Tables

**Figure 1 sensors-23-00617-f001:**
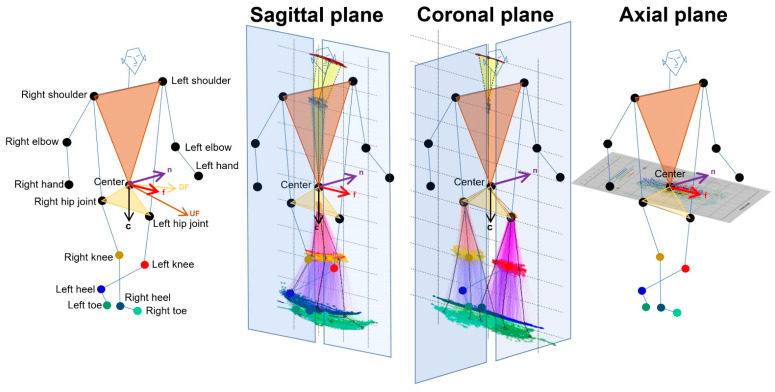
Projection onto the sagittal, coronal, and axial planes based on the body axis.

**Figure 2 sensors-23-00617-f002:**
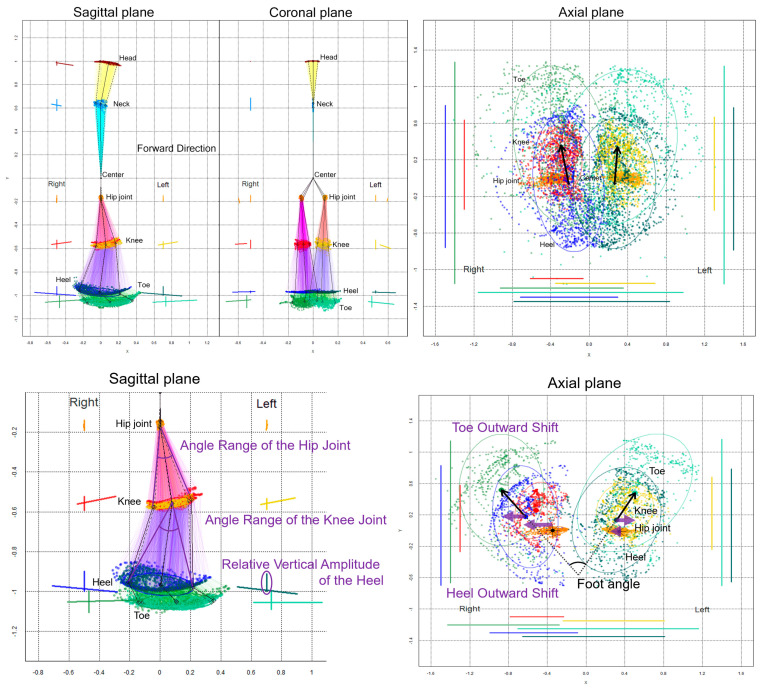
Two-dimensional relative coordinates projected onto the sagittal, coronal, and axial planes during two rounds of a 1-m diameter circular walk of a representative healthy volunteer. The chronological changes of the head (brown), neck (sky blue), right hip joint (orange), left hip joint (gold), right knee (red), left knee (yellow), right heel (blue), left heel (dark green), right toe (light green), and left toe (pale green) were plotted. The coordinates between joints, such as the hip and knee, knee and heel, and heel and toe, are connected by colored lines in the sagittal and coronal projection planes. In addition, 75% tolerance ellipses of all plots for each key point and its major axes and y-axis widths were drawn with each colored line on the left side of plots for the head and neck and the bilateral side of plots for legs. On the axial projection plane, plots and 75% tolerance ellipses for both legs and the horizontal and front-back range of motion for each key point were drawn with each colored line on the left, right, and bottom. The black arrows indicate vectors from the elliptical centers of the right and left hip joints to those of the knees.

**Figure 3 sensors-23-00617-f003:**
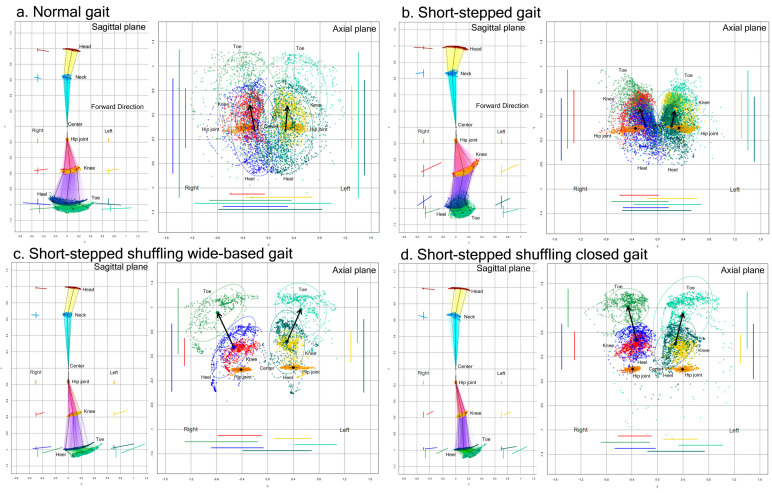
Two-dimensional relative coordinates projected onto the sagittal and axial planes of the body axis. A representative healthy volunteer performed a normal gait (**a**), short-stepped gait (**b**), short-stepped shuffling wide-based gait (**c**), and short-stepped shuffling closed gait (**d**) while the TDPT-GT application was recording. The colored coordinates and lines of the head (brown), neck (sky blue), right hip joint (orange), left hip joint (gold), right knee (red), left knee (yellow), right heel (blue), left heel (dark green), right toe (light green), and left toe (pale green) were plotted on the sagittal and axial projection planes.

**Figure 4 sensors-23-00617-f004:**
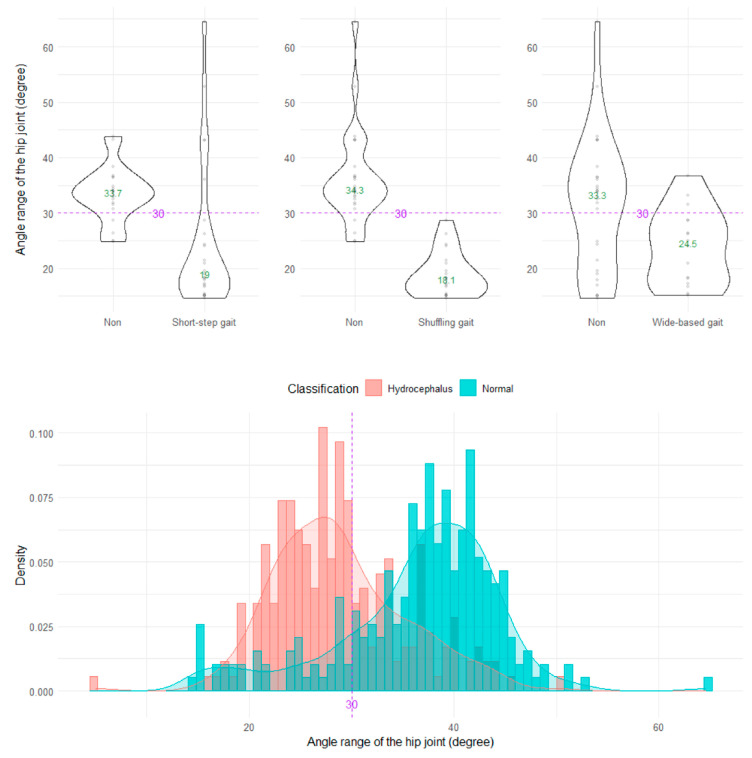
Distribution of angle ranges of the hip joint. The three violin plots in the upper row show the distribution of the angle range of the hip joint for the shuffling gait, short-stepped gait, and wide-based gait in the first 15 healthy volunteer group. The histogram in the lower row shows the distribution of the angle range of the hip joint for 47 patients with idiopathic normal pressure hydrocephalus (salmon pink) and 92 healthy elderly individuals aged 60 years or older in the Takahata cohort (blue emerald). The purple dotted line and number 30 indicate the proposed optimal cutoff value, and the green number in the violin plot indicates the median value.

**Figure 5 sensors-23-00617-f005:**
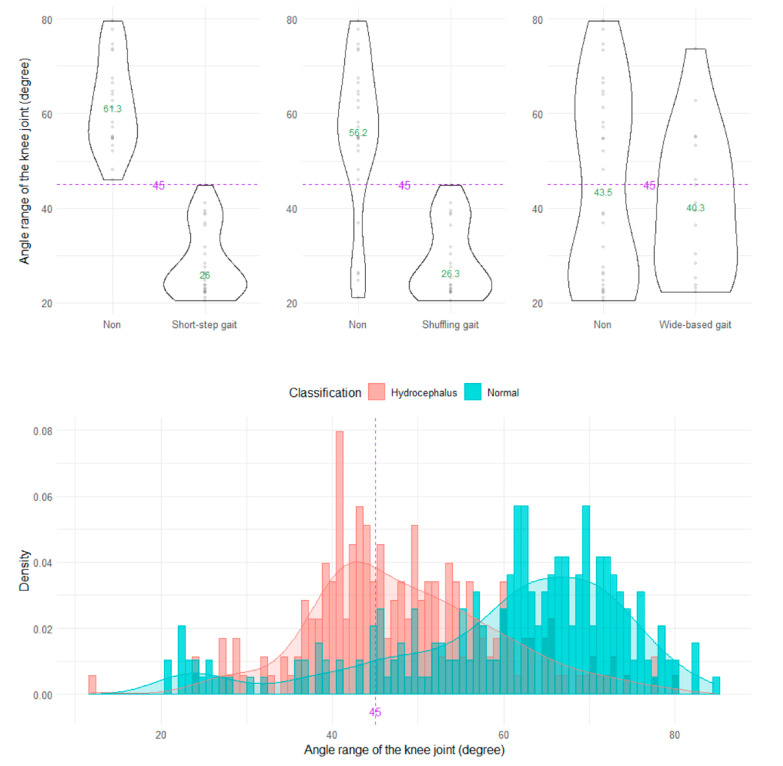
Distribution of angle ranges of the knee joint.

**Figure 6 sensors-23-00617-f006:**
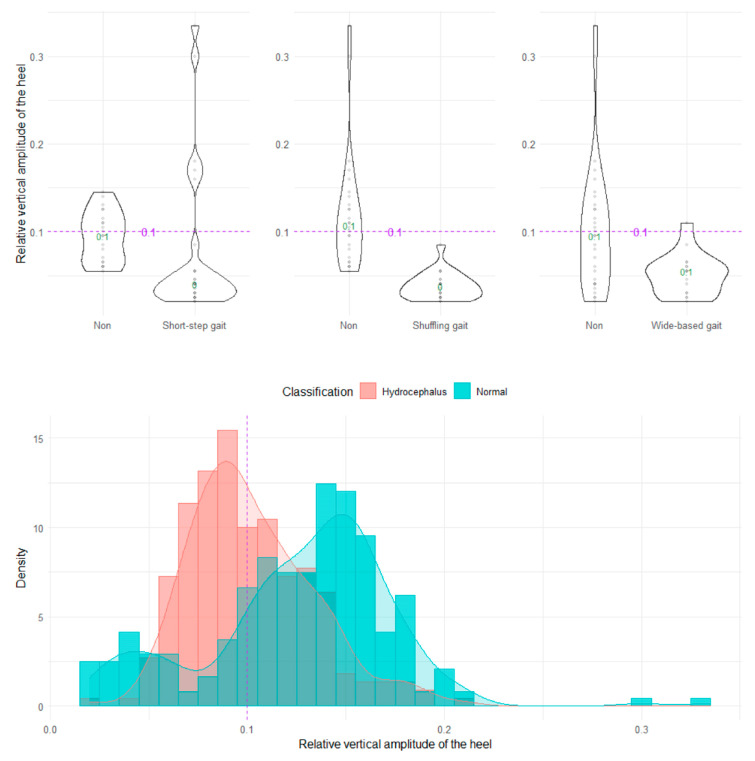
Distribution of relative vertical amplitudes of the heel.

**Figure 7 sensors-23-00617-f007:**
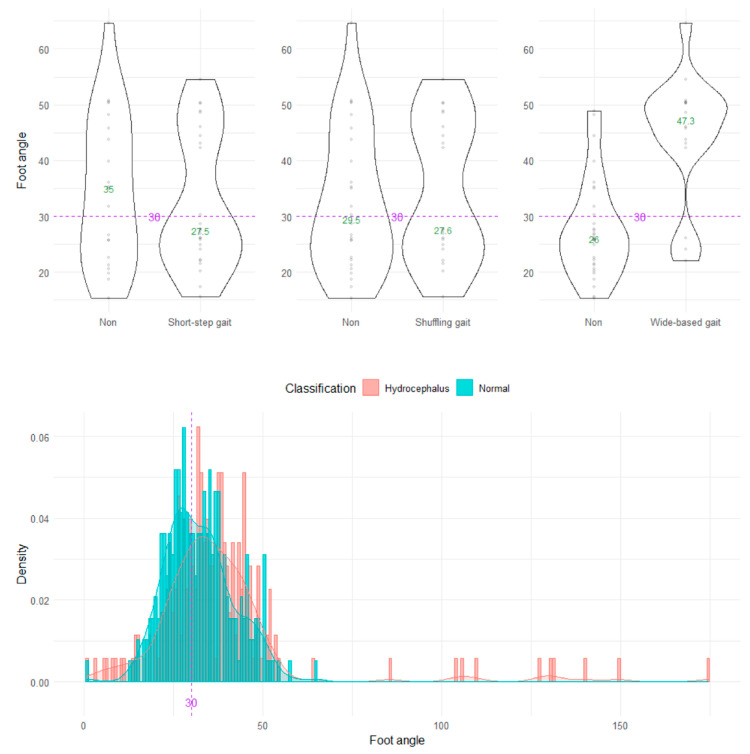
Distribution of foot angles. The three violin plots in the upper row show the distribution of the foot angle for the shuffling gait, short-stepped gait, and wide-based gait in the first 15 healthy volunteer group. The histogram in the lower row shows the distribution of the angle range of the hip joint for 47 patients with idiopathic normal pressure hydrocephalus (salmon pink) and 92 healthy elderly individuals aged 60 years or older in the Takahata cohort (blue emerald). The purple dotted line and number 30 indicate the proposed optimal cutoff value, and the green number in the violin plot indicates the median value.

**Figure 8 sensors-23-00617-f008:**
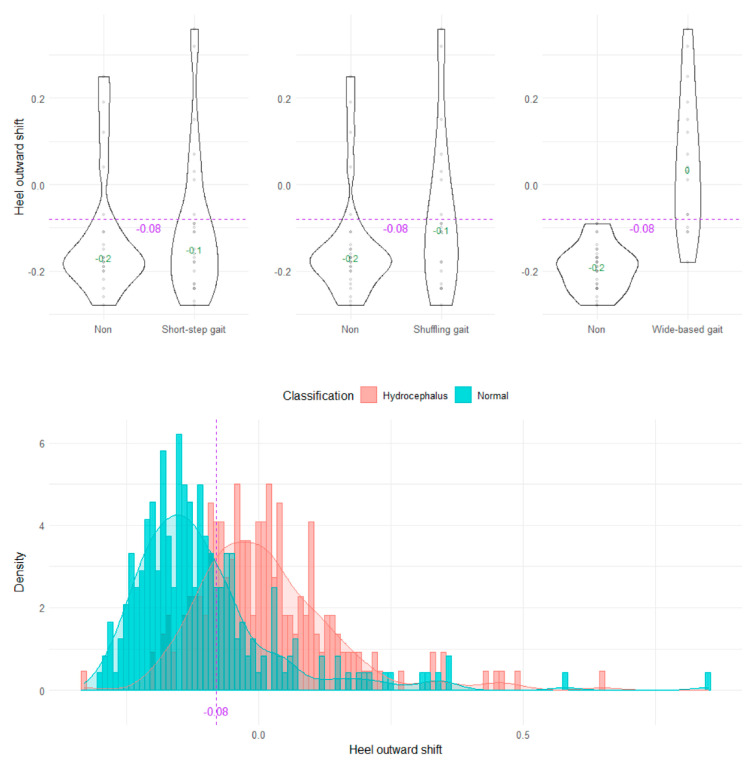
Distribution of heel outward shifts.

**Figure 9 sensors-23-00617-f009:**
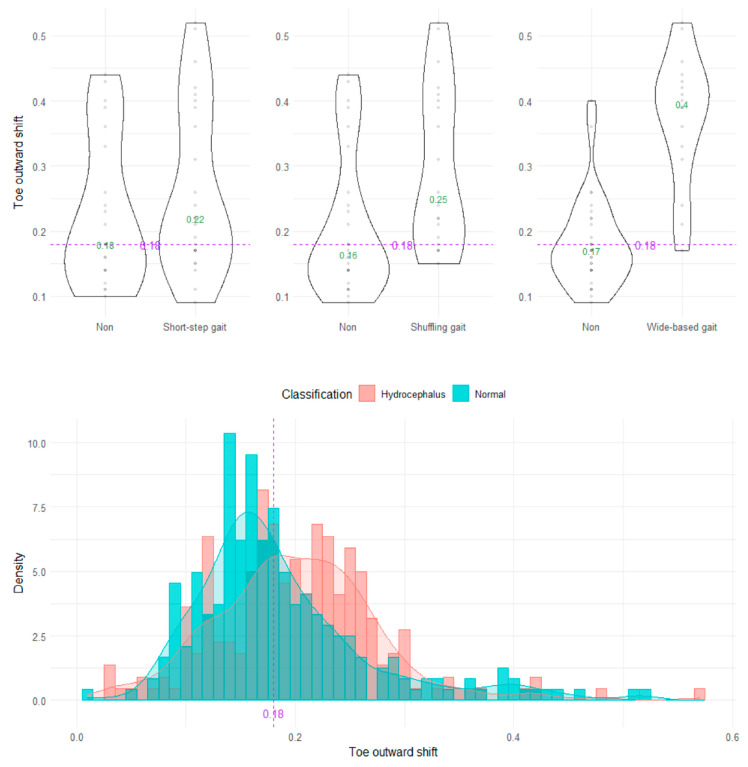
Distribution of toe outward shifts.

**Figure 10 sensors-23-00617-f010:**
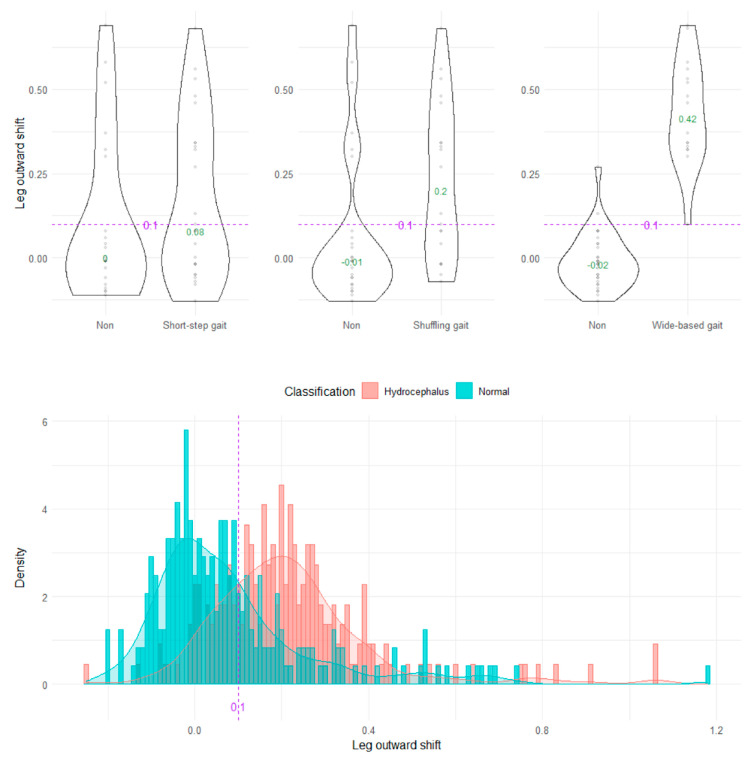
Distribution of leg outward shifts.

**Table 1 sensors-23-00617-t001:** Clinical characteristics of three datasets.

	Primary Healthy Volunteer	Takahata Cohort	iNPH Patient
Total Number	15	92	47
Sex (male:female)	9:6	36:56	32:15
Mean ± SD ^1^ of age (years)	39.1 ± 20.1	73.0 ± 6.3	77.3 ± 6.3
Range of age (years)	22–78	60–91	61–87
Shuffling gait	0	0	26
Short-stepped gait	0	0	38
Wide-based gait	0	0	35
Freezing gait	0	0	8
Spastic gait	0	0	2
Instability	0	0	41
Alzheimer’s disease	0	0	4
Fall history	0	19	30

^1^ SD: standard deviation.

**Table 2 sensors-23-00617-t002:** Detectability of a shuffling gait, short-stepped gait, and wide-based gait.

	Cutoff Value	AUC * (95% CI) ^”^	Sensitivity	Specificity	OD ^#^ (95% CI) ^”^
Shuffling gait					
Angle range of the hip joint (°)	30	77.1 (72.0–82.2)	69.0	74.3	6.39 (3.90–10.64)
Angle range of the knee joint (°)	45	78.6 (73.7–83.4)	52.6	87.5	7.71 (4.54–13.28)
Relative vertical amplitude of the heel	0.1	71.7 (66.3–77.2)	59.5	76.0	4.64 (2.87–7.57)
Short-stepped gait					
Angle range of the knee joint (°)	45	81.7 (77.4–86.0)	43.0	93.4	10.55 (5.63–21.12)
Short-stepped gait					
Heel outward shift	−0.08	78.1 (73.6–82.5)	81.4	60.9	3.36 (2.16–5.27)
Toe outward shift	0.18	69.8 (64.6–75.0)	70.2	58.8	6.77 (4.15–11.30)
Leg outward shift	0.1	80.6 (76.4–84.8)	81.4	63.8	7.65 (4.68–12.80)

* AUC: area under the receiver-operating characteristic curve. ^”^ 95% CI: 95% confidence interval. ^#^ OD: odds ratio.

## Data Availability

Data generated or analyzed during the study are available from the corresponding author upon reasonable request.

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
