# Peer review of "Quantitative Gait Feature Assessment on Two-Dimensional Body Axis Projection Planes Converted from Three-Dimensional Coordinates Estimated with a Deep Learning Smartphone App"

_sensors, 2023, doi:10.3390/s23020617_

Round 1
Reviewer 1 Report
27 Dec. 2021
Dear Authors,
Thank you very much for trusting me and inviting me to comment on this article. My comments on the article are as follows:
The authors designed an application using deep learning methods. This procedure estimates the relative 3D coordinates of 24 key points in the pose of the whole body during motion. A new method was also developed for easier assessment of pathological gait by converting 3D relative images into 2D coordinates projected relative to the body axis in sagittal, coronal and axial planes. This research has a very important role in the medical field. Based on the above considerations, I agree to recommend this paper for publication in your journal.
I am looking forward to hearing from you.
Sincerely yours,
Reviewer.
Author Response
Thank you for your adequate peer review.
Reviewer 2 Report
The study presented in the paper focuses on data collection and analysis using machine learning to evaluate pathological gait. The paper is well-written and easy to follow. The topic, the data collected and the approaches are relevant for bothe the research communities working in the field of pathological gait and the clinicians. The methodology is sound and the data sets are important. The results are clearly described and interesting. The authors themselves point a number of limitations, which is very good and appreciated.
The study will be useful for other researchers and cinicians, if published.
Author Response

(The authors gave the same response as above.)

Reviewer 3 Report
The authors described the application of an AI model that supports extraction of motion coordinates in 3D using a simple iphone camera. This research is extremely useful and if validated successfully in further larger cohorts can significantly improve usability of digital tools for gait assessment.
I have following few comments regarding the article:
What is the error in the AI model relative to the gold standard (3D motion capture) and how is this transmitted to gait features assessed in this study?
What is error in mapping 3D coordinates from the AI application to 2D relative coordinates in the sagittal, coronal and axial planes?
Can section 2.5 describing data analysis be expanded for additional detail, especially the steps involved and adequate references to the figures?
Was there any difference observed in pathological gait between acting it out and truly pathological population?
Was data for Takahata cohort and iNPH patient collected with TDPT-GT or a 3D motion capture system?
Is the AI model error assumed to be the same for populations with pathological and normal gaits? Is there any study or reference to validate robustness of the AI model for coordinate projection even in participants with pathological gaits?
Another minor comments, it would be great if the authors could explain all the figures in the manuscript in greater detail.
Author Response
Thank you for your instructive comments.
#Q1. What is the error in the AI model relative to the gold standard (3D motion capture) and how is this transmitted to gait features assessed in this study?
#R1. In our previous paper (Aoyagi et al. Sensors (Basel), 22, (14) 2022), we conducted the comparison between the novel markerless monocular motion capture smartphone application using deep learning model, named ‘TDPT for Gait Test (TDTP-GT)’ and the optical 3D motion capture systems with multipoint cameras and markers for motion capture, ‘VICON Motion System’, the gold standard for motion analysis systems.
However, the reliability and validity of the 3D relative coordinates estimated by the TDPT-GT have not yet been fully verified, that is, the validation results were still insufficient because the spatial coordinate system could not be transformed from the 3D relative coordinates on the TDTP-GT and global coordinate system on the VICON system. The 3D relative coordinates on the TDPT-GT were normalized as 1 for the length of the upper body from the navel to the center of the head, and 1 for the length of the lower body from the navel to the ankle, regardless of the subject's height. There are still challenges in verifying the accuracy of the TDPT-GT such as impossible to align the time by heel contact and different scales for the upper and lower body, but the validation study is ongoing in our study group. Therefore, we did not intend to publish them in this paper, and would present the validation results in the feature.
#Q2. What is error in mapping 3D coordinates from the AI application to 2D relative coordinates in the sagittal, coronal and axial planes?
#R2. The methods for converting 3D relative coordinates to 2D coordinates projected in the sagittal, coronal, and axial planes relative to the body axis are basically the same whether using the TDPT-GT or the VICON system, and no specific errors would exist.
#Q3. Can section 2.5 describing data analysis be expanded for additional detail, especially the steps involved and adequate references to the figures?
#R3. According to the reviewer's advice, we have added an explanation of the 75% tolerance (confidence) ellipse calculated by using dataEllipse in the R car package, as follow: dataEllipse(x coordinates of key point, y coordinates of key point, levels = 0.75). The center coordinate of the ellipse was calculated as cov.wt(cbind(x coordinates of key point, y coordinates of key point)), and the elliptical area was calculated as the length of major axis of ellipse × the length of minor axis of ellipse × π (pi).
In addition, we have added our newly defined indicators into new Figure 2.
A relative vertical amplitude of the heel was defined as the y-axis width of the heel plots.
#Q4. Was there any difference observed in pathological gait between acting it out and truly pathological population?
#R4. As described in the Discussion section, the pathological gait patterns of the shuffling, short-stepped, and wide-based gaits are closely related to each other and could not be clearly distinguished in the real patients with iNPH and older adults. Therefore, we needed to evaluate the simulated pathological gaits in volunteers at first. The simulated pathological gaits of the healthy volunteers were quite extreme, while the iNPH patients often had several degrees of pathological gaits, but it was difficult to maintain consistency in their evaluation. Therefore, it is easy to polarize the parameters in the simulated pathological gaits in the healthy volunteers, and it is easy to find candidate parameters, indicators, and their cutoff values related to the pathological gaits.
We have added one sentence in the 2.2. Study population.
#Q5. Was data for Takahata cohort and iNPH patient collected with TDPT-GT or a 3D motion capture system?
#R5. Of course, all subjects, including dataset from the iNPH patients and Takahata cohort, had gait data collected with TDPT-GT. Nowhere in this paper is it described that any 3D motion capture systems were used, and it is clear that gait data was collected with TDPT-GT.
#Q6. Is the AI model error assumed to be the same for populations with pathological and normal gaits?
#R6. This paper is not about developing the TDTP-GT deep learning smartphone application. It was described in detail in the paper that precedes this paper in Sensors (Basel) (Aoyagi et al. Development of smartphone application for markerless three-dimensional motion capture based on deep learning model. 2022). Nor is this a study of an AI model to determine normal or pathological gait. It is unlikely that there would be a difference in the 3D coordinates or 2D projected coordinates collected for normal and pathological gait patterns.
#Q7. Is there any study or reference to validate robustness of the AI model for coordinate projection even in participants with pathological gaits?
#R7. We developed a new method for assessing pathological gaits more easily by converting 3D relative coordinates into 2D coordinates projected on the sagittal, coronal, and axial planes with respect to the body axis, as described in the Introduction section.
This is a completely new method and, to our knowledge, no similar work has been published to date and no references will exist.
As mentioned in the first limitation, the 2D relative coordinates projected onto the sagittal or axial planes of the body axis based on the navel as the center without the ground information have not been compared with the previous conventional gait research.
#Q8. It would be great if the authors could explain all the figures in the manuscript in greater detail.
#R8. Many of the explanations for the figures are provided in the text, and we have taken care to avoid overlapping the same explanations. However, we made the explanations as detailed as possible, according to the reviewer's advice.